# PeerJ

# Honey bee success predicted by landscape composition in Ohio, USA

DB Sponsler and RM Johnson

Department of Entomology, The Ohio State University, Wooster, OH, USA

## ABSTRACT

Foraging honey bees (*Apis mellifera* L.) can routinely travel as far as several kilometers from their hive in the process of collecting nectar and pollen from floral patches within the surrounding landscape. Since the availability of floral resources at the landscape scale is a function of landscape composition, apiculturists have long recognized that landscape composition is a critical determinant of honey bee colony success. Nevertheless, very few studies present quantitative data relating colony success metrics to local landscape composition. We employed a beekeeper survey in conjunction with GIS-based landscape analysis to model colony success as a function of landscape composition in the State of Ohio, USA, a region characterized by intensive cropland, urban development, deciduous forest, and grassland. We found that colony food accumulation and wax production were positively related to cropland and negatively related to forest and grassland, a pattern that may be driven by the abundance of dandelion and clovers in agricultural areas compared to forest or mature grassland. Colony food accumulation was also negatively correlated with urban land cover in sites dominated by urban and agricultural land use, which does not support the popular opinion that the urban environment is more favorable to honey bees than cropland.

## INTRODUCTION

Honey bees (*Apis mellifera*, L.) exist in large, eusocial colonies that require massive and sustained inputs of floral nectar and pollen. They meet this demand by foraging at an extremely large spatial scale and with rapid responsiveness to changes in the surrounding floral community (*Visscher & Seeley, 1982*; *Seeley, 1995*). Depending on local floral availability, colonies may routinely forage over an area of more than 100 km$^2$ (*Seeley, 1995*), and much larger ranges have been reported under extreme conditions (*Eckert, 1931*; *Beekman & Ratnieks, 2001*).

Because honey bee foraging is a decidedly landscape-scale process, one should expect landscape composition to interact meaningfully with colony nutrition and overall colony success. While the plausibility of such a relationship is widely acknowledged (*Steffan-Dewenter & Kuhn, 2003*; *Naug, 2009*; *Van Engelsdorp & Meixner, 2010*; *Härtel & Steffan-Dewenter, 2014*), and the importance of apiary location is axiomatic among practicing beekeepers, there are very few published studies that quantitatively measure

Corresponding author
DB Sponsler, sponsler.18@osu.edu

colony success in response to local landscape variables. As rapid landscape conversion continues as a global phenomenon, and beekeepers in many regions continue to suffer unsustainable losses, the task of refining and expanding our knowledge of honey bee landscape ecology takes on obvious urgency.

Several studies have indirectly explored the relationship between landscape and colony success by analyzing the spatial information encoded in the honey bee dance language (*von Frisch , 1967*). *Waddington et al. (1994)* found that colonies located in two suburban landscapes tended to forage over a smaller area and with a less clumped distribution than a previously studied colony located in a temperate deciduous forest (*Visscher & Seeley, 1982*), suggesting that suburban landscapes might provide richer and more evenly distributed resource patches. Similarly, *Garbuzov, Schürch & Ratnieks (2014)* found that colonies in the city of Brighton, UK, concentrated most of their foraging within city limits rather than venturing into surrounding countryside that was well within their foraging range. Conversely, *Beekman & Ratnieks (2001)* observed remarkably long-distance foraging under conditions of apparently scarce local resources in a suburban landscape and highly rewarding resources in outlying seminatural heather moors. In agricultural landscapes, honey bee foraging patterns suggest that pollen sources can be scarcer and floral patches less spatially and temporally variable in highly simplified cropping systems compared to more structurally complex habitats (*Steffan-Dewenter & Kuhn, 2003*), while conservation management within farmlands can increase the availability of bee-attractive flora (*Couvillon, Schürch & Ratnieks, 2014*).

Landscape composition can also influence the type and quality of pollen foraged by honey bees. *Donkersley et al. (2014)* found that the protein content of "beebread" (processed pollen stored by honey bees) was negatively correlated with agricultural land cover and positively correlated with broad-leaf forest, improved grassland, and urban land cover.

Two recent studies have directly related colony success to local landscape variables (*Sande et al., 2009*; *Odoux et al., 2014*). In the dry coastal forest habitat of southeastern Kenya, *Sande et al. (2009)* found that a colony's honey production was positively correlated with its proximity to forest patches. *Odoux et al. (2014)* similarly found that colony size was positively correlated with forest land cover in the intensively agricultural landscape of central-western France.

Among non-peer-reviewed sources, there is a widely circulated opinion that honey bee success is favored by urban/suburban landscapes, especially in comparison to cropland (*Graham, 1992*; *Anonymous, 2008*; *Wilson-Rich, 2012*). These claims remain unsubstantiated but plausible given the ostensibly positive effects of suburban land use suggested by *Waddington et al. (1994)* and the more direct evidence supporting the favorability of suburban land use for bumble bees (Hymenoptera: *Bombus*, Latreille) living in predominantly agricultural areas (*Goulson et al., 2002*; *Goulson et al., 2010*).

Here, we present a quantitative study of honey bee colony success in relation to landscape composition in the State of Ohio, USA, a region characterized by a mixture of intensive cropland, deciduous forest, grassland, and urban development. While there are many ways to measure colony success, we focus on four metrics that are highly

 

relevant to beekeepers and easily assessed through simple hive inspection: honey and pollen accumulation, wax production, adult population, and brood population. Using a citizen-science survey, we investigate the relationship between colony success and the landscape as a whole, accounting for all major land cover types and also for the potential influence of hive management variables that vary between beekeepers. Then, we specifically evaluate the putative favorability of urban land use using a subset of sites dominated by urban development and/or cropland.

## MATERIALS AND METHODS

### Survey design

In 2012 and 2013, we used a survey-based, citizen-science approach to measure the productivity of honey bee colonies in the state of Ohio, USA. All participants were beekeepers whose hives were registered with the Ohio Department of Agriculture and who volunteered to participate in our study. Volunteers were enlisted through a combination of email communications, public speaking engagements, and cooperation with local beekeeping organizations; our study was publicized as widely as possible, and we did not attempt to target any particular demographic. Our survey was conducted with written exemption from IRB review by the Ohio State University Office of Responsible Research Practices (Protocol # 2012E0136 and 2013E0012).

In order to standardize the initial strength of the colonies in our study (hereafter "study colonies") and minimize the influence of parasites and pathogens, we restricted our study to colonies that had been started from artificial swarms, known as "package bees," in the spring of each study year. Honey bee packages are created by combining a standard quantity of worker bees (usually 1.36 kg) with a newly mated queen. The initial strength of colonies started from package bees is, therefore, less variable than that of over-wintered colonies. Moreover, because they are sold without comb or brood, they tend to have reduced parasite and pathogen loads.

Data for each study colony were gathered using a two-part survey consisting of spring and fall components (hereafter "spring survey" and "fall survey"). The spring survey was made available beginning in early March, and participants were instructed to complete the survey immediately after installing their honey bee packages. In the spring survey, we gathered the geographic location of each study colony and the years of experience of each participating beekeeper (see Supplemental Information S1 for full spring survey questionnaire). The fall survey was made available in mid-September and completed by mid-October. To complete the fall survey, each participant performed a frame-by-frame hive inspection and reported the number of frames in the study hive belonging to the following categories: (1) more than half honey/nectar, (2) more than half pollen, (3) more than half brood, (4) more than half empty wax comb, (5) more than half bare foundation (no wax comb). Participants also reported the quantity of sugar syrup that had been given to their hives as supplemental feeding, a common beekeeping practice that could affect colony success. See Supplemental Information S2 for the full fall survey questionnaire.

## Survey processing

Each beekeeper was instructed to submit data for only one study hive at one apiary site, and each beekeeper was included in only one of the two years of our study. The data quality of each survey was carefully vetted prior to analysis, and surveys missing critical data or having irreconcilable inconsistencies were discarded. Fall surveys reporting hives that had died since spring installation were also discarded. The final numbers of surveys included in analyses for 2012 and 2013 were 32 and 18, respectively; these were selected from a pre-processing total of 55 surveys in 2012 and 33 in 2013. The minimum distance between study hives, combining both years, was 2.65 km.

From our survey data, we derived four metrics to represent colony success: *net food accumulation*, *net wax production*, *adult population*, and *brood population*. For consistency, all metrics were recorded in units of standard deep frames.
*Net food accumulation*:

$$Food = H + H_{\text{harv}} - H_{\text{add}} + P$$

where $H$ = honey/nectar frames in hive at time of inspection, $H_{\text{harv}}$ = honey frames harvested prior to inspection, $H_{\text{add}}$ = honey frames added to the hive prior to inspection (beekeepers sometimes transfer honey frames between hives to increase food stores of weak colonies), and $P$ = frames of pollen in hive at time of inspection. This variable will hereafter be abbreviated *Food*.
*Net wax production*:

$$Wax = H + H_{\text{harv}} + P + B + B_{rm} + D - H_{\text{add}} - B_{\text{add}} - D_{\text{add}}$$

where $B$ = brood frames in hive at time of inspection, $B_{rm}$ = brood frames removed prior to inspection (brood frames may be transferred between colonies to modulate population size), $D$ = drawn but mostly empty frames in hive at time of inspection, $B_{\text{add}}$ = brood frames added to the hive prior to inspection, and $D_{\text{add}}$ = drawn but mostly empty frames (frames with wax comb constructed but no cell contents) added to hive prior to inspection. This variable will hereafter be abbreviated *Wax*.

*Adult population* (hereafter, *AdultPop*) was measured as the number of frames "more than half covered" with adult bees at time of inspection. *Brood population* (hereafter, *BroodPop*) was simply the number of "mostly brood" frames reported by the inspecting beekeeper.

We also measured two hive management variables: years of beekeeping experience of the participating beekeeper (*experience*) and quantity of sugar syrup fed to the study hive since its installation (*syrup*).

## Landscape analysis

Geographic coordinates for each study hive were determined and mapped using QGIS v.2.1 (*QGIS Development Team , 2014*). To encompass a range of spatial scales at which landscape effects on colony success might be seen, we defined the landscape of each hive

using six nested buffers having radii of 0.5, 1, 2, 3, 4, and 5 km, respectively. Land cover data for the State of Ohio were obtained from the 2006 dataset provided by the National Land Cover Database (NLCD 2006) (*Fry et al., 2011*). The NLCD 2006 land cover layer for Ohio is comprised primarily of seven land cover classes: *cultivated crops, pasture/hay, deciduous forest*, and four levels of urban development (*open space, low intensity, medium intensity, high intensity*). Minor classes, present only at very low abundance, include *evergreen forest, mixed forest, woody wetland, herbaceous wetland, grassland/herbaceous, shrub/scrub, barren land*, and *open water*. To simplify our analysis of landscape composition, we condensed the non-crop land cover classes (ignoring *barren land* and *open water*) into three aggregate classes: *Forest* (*deciduous + evergreen + mixed + woody wetland + shrub/scrub*), *Grassland* (*pasture/hay + grassland/herbaceous + herbaceous wetland*), and *Urban* (*open space + low intensity + medium intensity + high intensity*). The landscape composition of each study site, measured in terms of the total land cover of *Crop* (*cultivated crop*) and each aggregate class, was determined at each spatial scale using LECOS (Jung, 2013), a QGIS plugin for calculating patch-based landscape metrics. As a measure of overall landscape heterogeneity, we also calculated Simpson's Diversity Index (*D*) based on the original, non-aggregated land cover classes.

## Data analysis

We first reduced the dimensionality of our landscape data using principal components analysis (PCA) based on the covariance between the variables *Crop, Forest, Grassland*, and *Urban*. This step was repeated for each spatial scale. For all scales, the first two principal components (PC1 and PC2) explained >96% of total variance.

To model the relationship between landscape composition and colony success, accounting also for the management variables *experience* and *syrup*, we conducted model selection using Akaike's Information Criterion corrected for small sample size ($AIC_c$) (*Burnham & Anderson, 2002*). Each success metric–*Food, Wax, AdultPop*, and *BroodPop*–was modeled separately. Fourteen candidate linear models were constructed for each success metric at each spatial scale; these included all combinations of the landscape variables (PC1, PC2, *D*) and the coupled management variables *experience* and *syrup*, a year-only model, and an intercept-only model. For each success metric, we present the candidate model having the lowest $AIC_c$ score at each scale along with any competing models having an $AIC_c$ difference of <2 (Table 1) (*Burnham & Anderson, 2002*). We then selected a single best model for each success metric by choosing the model with the lowest $AIC_c$ score across all spatial scales.

To evaluate the prediction that urban land cover favors honey bee success relative to agricultural land cover, we first extracted the subset of our sites ($n = 30–33$, varying with spatial scale) for which *Urban + Crop* was greater than 50% of total landcover, a threshold chosen *a priori* to identify sites that were strongly characterized by urban and/or agricultural land use. Then, we then set up separate linear regression models for *Food* and *Wax* with *Urban* as the explanatory variable. Only *Food* and *Wax* were analyzed because the results of the PCA described above indicated that only these two success metrics should

Sponsler and Johnson (2015), *PeerJ*, DOI 10.7717/peerj.838

**Table 1 Summary of model selection statistics for each colony success metric.** Only models with AIC$_c$ < 2 are presented as competing models. Models within each spatial scale are listed in order of increasing AICc value. The best model for each success metric is depicted in bold.

| Metric | Radius (km) | Model | Log-likelihood | $K_i$ | AIC$_c$ | $\Delta$AIC$_c$ | $W_i$ | Adjusted $r^2$ | Coefficients |
|---|---|---|---|---|---|---|---|---|---|
| Food | 0.5 | PC2 | −165.808 | 3 | 338.138 | 0.00 | 0.233 | 0.047 | −5.9142 |
| " | 0.5 | PC1 + PC2 | −165.060 | 4 | 339.008 | 0.87 | 0.151 | 0.055 | PC2 = −5.9142, PC1 = 2.5032 |
| " | 0.5 | Intercept | −167.515 | 2 | 339.286 | 1.15 | 0.563 | 0.131 | |
| **"** | **1** | **PC2** | **−165.134** | **3** | **336.791** | **0.00** | **0.260** | **0.072** | **−7.3139** |
| " | 1 | PC1 + PC2 | −164.175 | 4 | 337.240 | 0.45 | 0.208 | 0.088 | PC2 = −7.3139, PC1 = 2.9608 |
| " | 2 | PC2 | −165.686 | 3 | 337.894 | 0.00 | 0.197 | 0.051 | −6.541 |
| " | 2 | PC1 + PC2 | −164.553 | 4 | 337.995 | 0.10 | 0.187 | 0.074 | PC2 = −6.5409, PC1 = 3.5536 |
| " | 2 | intercept | −167.515 | 2 | 339.286 | 1.39 | 0.499 | 0.098 | |
| " | 2 | PC1 + PC2 + D | −163.990 | 5 | 339.343 | 1.45 | 0.095 | 0.075 | PC2 = −7.529, PC1 = 5.195, D = 7.674 |
| " | 2 | PC1 | −166.464 | 3 | 339.450 | 1.56 | 0.090 | 0.021 | 3.5536 |
| " | 3 | PC2 | −165.871 | 3 | 338.265 | 0.00 | 0.183 | 0.044 | −6.0981 |
| " | 3 | PC1 + PC2 | −164.733 | 4 | 338.355 | 0.09 | 0.175 | 0.067 | PC2 = −6.0981, PC1 = 3.7970 |
| " | 3 | intercept | −167.515 | 2 | 339.286 | 1.02 | 0.600 | 0.110 | |
| " | 3 | PC1 | −166.451 | 3 | 339.424 | 1.16 | 0.103 | 0.022 | 3.7970 |
| " | 3 | PC1 + PC2 + D | −164.247 | 5 | 339.858 | 1.59 | 0.083 | 0.065 | PC2 = −6.554, PC1 = 5.729, D = 7.200 |
| " | 4 | PC2 | −166.135 | 3 | 338.791 | 0.00 | 0.179 | 0.034 | −5.5831 |
| " | 4 | intercept | −167.515 | 2 | 339.286 | 0.49 | 0.781 | 0.140 | |
| " | 4 | PC1 + PC2 | −165.202 | 4 | 339.293 | 0.50 | 0.139 | 0.050 | PC2 = −5.5831, PC1 = 3.5906 |
| " | 4 | PC1 | −166.634 | 3 | 339.789 | 1.00 | 0.109 | 0.015 | 3.5906 |
| " | 5 | PC2 | −166.203 | 3 | 338.928 | 0.00 | 0.174 | 0.031 | −5.378 |
| " | 5 | intercept | −167.515 | 2 | 339.286 | 0.36 | 0.836 | 0.145 | |
| " | 5 | PC1 + PC2 | −165.269 | 4 | 339.428 | 0.50 | 0.135 | 0.047 | PC2 = −5.3783, PC1 = 3.6745 |
| " | 5 | PC1 | −166.630 | 3 | 339.782 | 0.85 | 0.113 | 0.015 | 3.6745 |
| Wax | 0.5 | PC2 | −180.163 | 3 | 366.848 | 0.00 | 0.242 | 0.041 | −7.525 |
| " | 0.5 | Year | −180.538 | 3 | 367.598 | 0.75 | 0.687 | 0.167 | 0.02648 |
| " | 0.5 | intercept | −181.724 | 2 | 367.704 | 0.86 | 0.652 | 0.158 | |
| " | 1 | PC2 | −179.240 | 3 | 365.001 | 0.00 | 0.299 | 0.076 | −9.917 |
| " | 1 | PC2 + D | −178.958 | 4 | 366.804 | 1.80 | 0.122 | 0.067 | PC2 = −8.892, D = −6.540 |
| **"** | **2** | **PC2** | **−178.695** | **3** | **363.911** | **0.00** | **0.341** | **0.096** | **−11.053** |
| " | 2 | PC2 + D | −178.388 | 4 | 365.665 | 1.75 | 0.142 | 0.088 | PC2 = −10.247, D = −6.265 |
| " | 2 | PC2 + years + syrup | −177.249 | 5 | 365.862 | 1.95 | 0.129 | 0.109 | PC2 = −11.8583, years = 0.1252, syrup = 0.2578 |
| " | 3 | PC2 | −179.076 | 3 | 364.673 | 0.00 | 0.278 | 0.082 | −10.183 |
| " | 3 | PC2 + D | −178.374 | 4 | 365.636 | 0.96 | 0.172 | 0.088 | PC2 = −9.611, D = −9.020 |
| " | 3 | PC2 + years + syrup | −177.453 | 5 | 366.270 | 1.60 | 0.125 | 0.102 | PC2 = −11.4033, years = 0.1346, syrup = 0.2765 |
| " | 4 | PC2* | −179.411 | 3 | 365.344 | 0.00 | 0.260 | 0.069 | −9.514 |

Sponsler and Johnson (2015), *PeerJ*, DOI 10.7717/peerj.838

Table 1 (*continued*)

| Metric | Radius (km) | Model | Log-likelihood | $K_i$ | $AIC_c$ | $\Delta AIC_c$ | $W_i$ | Adjusted $r^2$ | Coefficients |
|---|---|---|---|---|---|---|---|---|---|
| " | 4 | PC2 + D | −178.721 | 4 | 366.331 | 0.99 | 0.159 | 0.075 | PC2 = −9.281, D = −8.998 |
| " | 4 | PC2 + years + syrup | −177.827 | 5 | 367.017 | 1.67 | 0.113 | 0.089 | PC2 = −10.7781, years = 0.1244, syrup = 0.2762 |
| " | 5 | PC2* | −179.465 | 3 | 365.451 | 0.00 | 0.255 | 0.067 | −9.290 |
| " | 5 | PC2 + D | −178.750 | 4 | 366.389 | 0.94 | 0.159 | 0.074 | PC2 = −9.253, D = −9.112 |
| " | 5 | PC2 + years + syrup | −177.865 | 5 | 367.095 | 1.64 | 0.112 | 0.087 | PC2 = −10.5842, years = 0.1317, syrup = 0.2776 |
| AdultPop | 2 | PC2 + years + syrup | −160.864 | 5 | 333.092 | 1.47 | 0.205 | 0.172 | PC2 = −3.0878, years = 0.4904, syrup = 0.2896 |
| " | 3 | PC2 + years + syrup | −160.590 | 5 | 332.544 | 0.92 | 0.247 | 0.181 | PC2: −3.7837, years: 0.4939, syrup: 0.2991 |
| " | 4 | PC2 + years + syrup | −160.652 | 5 | 332.668 | 1.05 | 0.235 | 0.179 | PC2 = −3.6243, years = 0.4906, syrup = 0.2993 |
| " | 4 | D + years + syrup | −161.090 | 5 | 333.544 | 1.92 | 0.151 | 0.164 | D: 4.2943, years: 0.5219, syrup: 0.3059 |
| " | 5 | PC2 + years + syrup | −160.634 | 5 | 332.631 | 1.01 | 0.234 | 0.180 | PC2 = −3.6267, years = 0.4931, syrup = 0.3002 |
| " | 5 | D + years + syrup | −161.002 | 5 | 333.367 | 1.75 | 0.162 | 0.167 | D = 4.9270, years = 0.5308, syrup = 0.3094 |
| **"** | **NA** | **years+syrup** | **−161.365** | **4** | **331.620** | **0.00** | **0.475** | **0.173** | **years = 0.4887, syrup = 0.2774** |

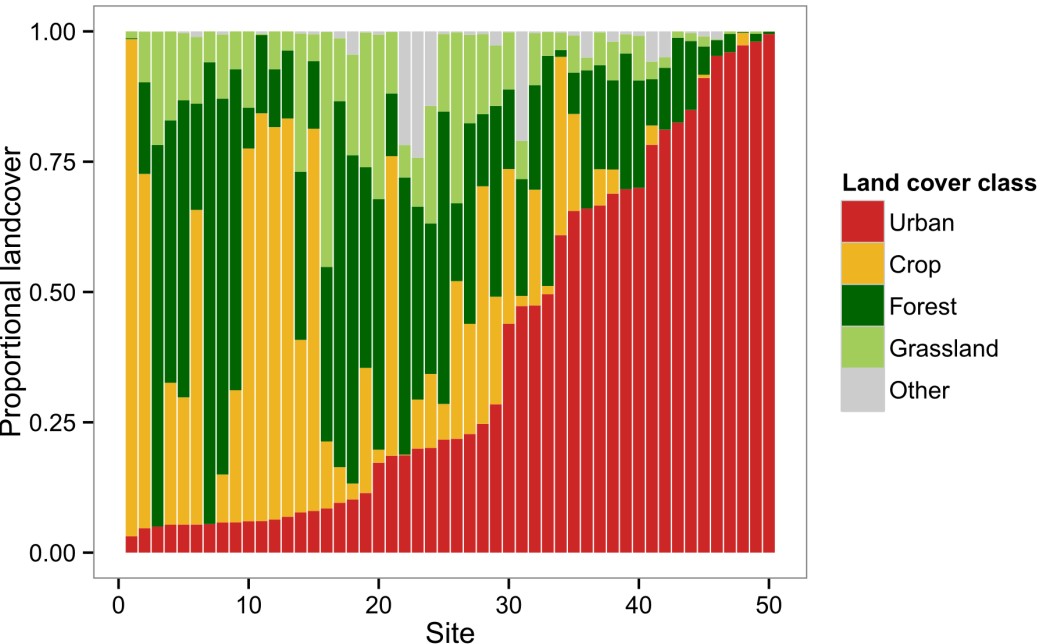

**Figure 1 Landscape composition of study sites at 2 km radius.** Sites are depicted in order of increasing urban (red) land cover. Other major land cover classes include crop (gold), forest (dark green), and grassland (light green). Remaining land cover (grey) consisted of barren land and open water.

be expected to respond to landscape variables. We did not use *experience* and *syrup* as covariates because previous analysis showed they were not predictive of *Food* or *Wax*. Regression analysis was repeated for each spatial scale.

All analyses were performed in R statistical software (*R Core Team, 2013*). AIC$_c$ model selection used the package `AICcmodavg` (Mazerolle, 2014). Model assumptions were verified by visual assessment using the `plot (lm)` function in R.

## RESULTS

### Landscape analysis

The landscapes surrounding the colonies in our survey represented a broad range of landscape composition in terms of the major land cover classes: *Crop*, *Forest*, *Grassland*, and *Urban* (Fig. 1). Principal components analysis of these four variables yielded two readily interpretable axes that explained greater than 96% of total variance (Fig. 2). PC1 was essentially an urban-rural axis, with sites dominated by *Urban* scoring low and sites dominated by combinations of *Crop*, *Forest*, and/or *Grassland* scoring high. PC2 partitioned non-urban landscapes into those characterized by *Crop* and those characterized by *Forest* and, to a lesser extent, *Grassland*.

### Modeling colony success metrics by landscape principal components

*Food* and *Wax* were best modeled with PC2 as the only explanatory variable. Almost all competing models ($\Delta$AIC$_c$ <2) included PC2 alongside other explanatory variables,

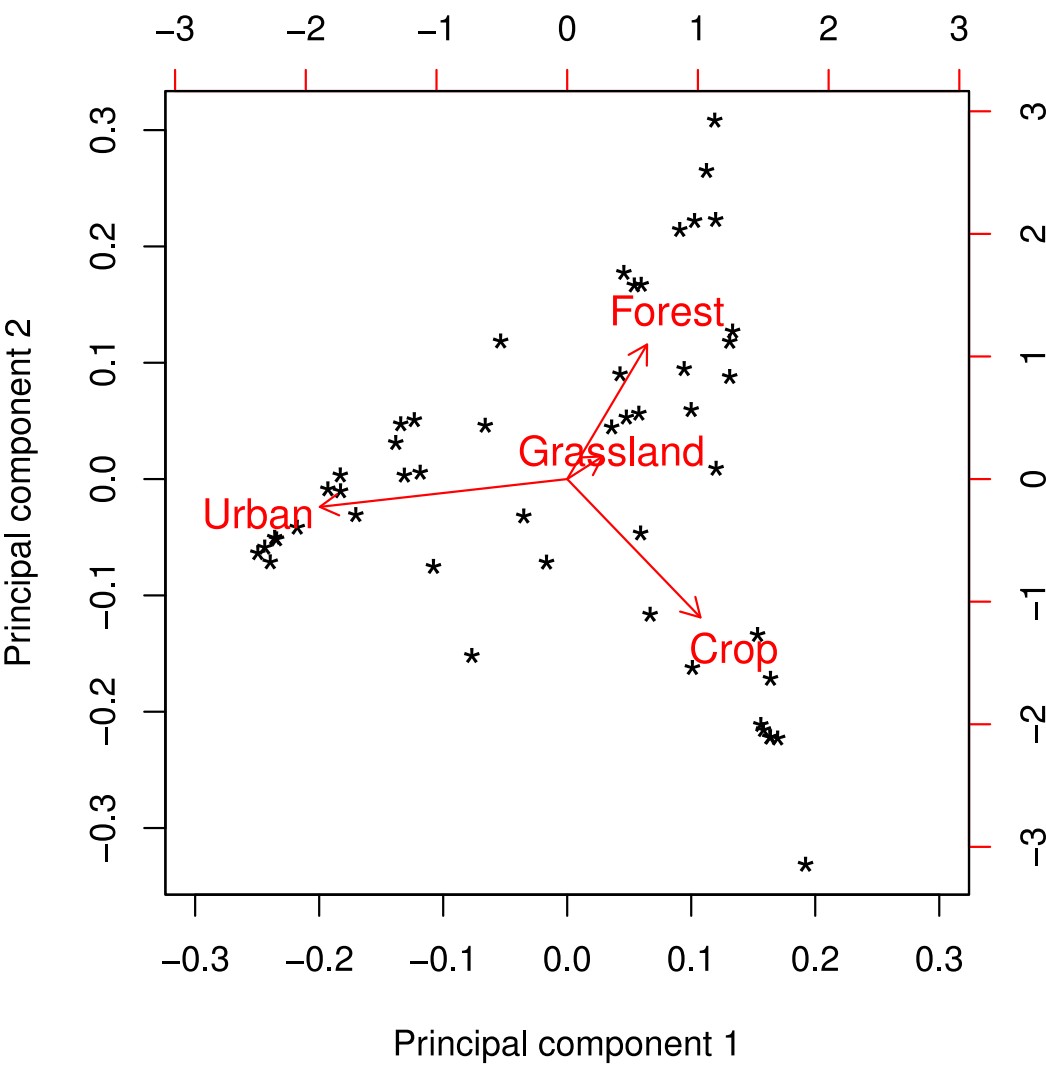

**Figure 2 Principal components biplot of major land cover classes at a radius of 2 km.** Principal component 1 (PC1) comprises an urban-rural axis, with lower scores corresponding to higher urbanness. Principal component 2 (PC2) forms an axis that separates sites characterized by forest/grassland from those characterized by cropland. This pattern was consistent at all spatial scales with only minor variation.

further supporting the conclusion that PC2 was the single most important predictor (Table 1). For *Food*, the optimal spatial scale was a 1 km radius, while *Wax* was best predicted at a 2 km radius. In both cases, the relationship was negative and the linear regression models were statistically significant (*Food*: $F = 4.796$, df = 48, $p = 0.033$; *Wax*: $F = 6.184$, df = 48, $p = 0.016$) (Fig. 3). *AdultPop* was best modeled with the coupled management variables *experience* and *syrup* as the only explanatory variables. The relationship was positive and the linear regression model was significant ($F = 6.128$, df = 47, $p = 0.004$), with significant contributions from both *experience* ($t = 2.98$, df = 47, $p = 0.005$) and *syrup* ($t = 2.474$, df = 47, $p = 0.017$) (Fig. 4). *BroodPop* was best predicted by the intercept-only model, indicating that none of our measured explanatory variables were good predictors of this success metric.

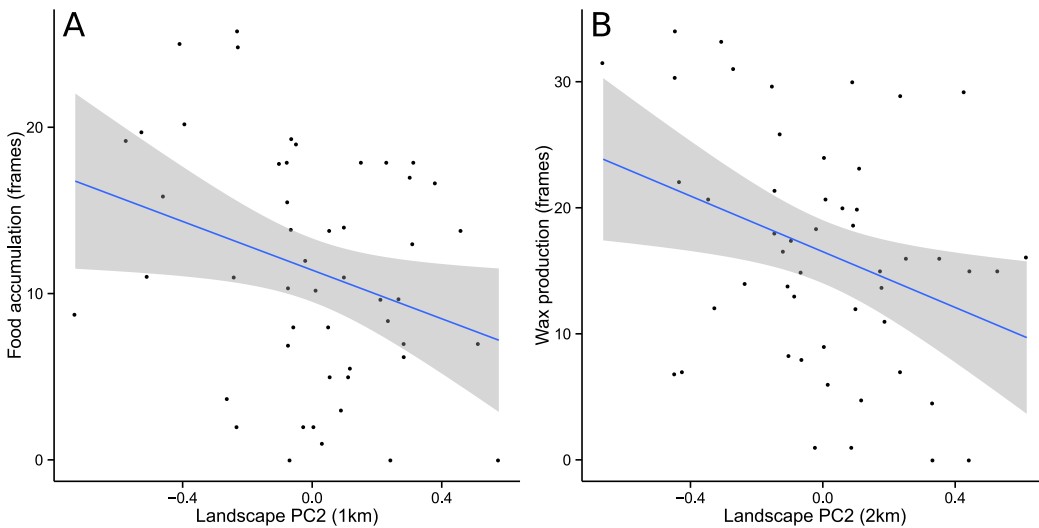

**Figure 3** **Food accumulation and wax production were negatively correlated with PC2.** This indicates that productivity in terms of food and wax increased in the direction of cropland and decreased in the direction of forest/grassland. A 95% confidence band is shaded in gray.

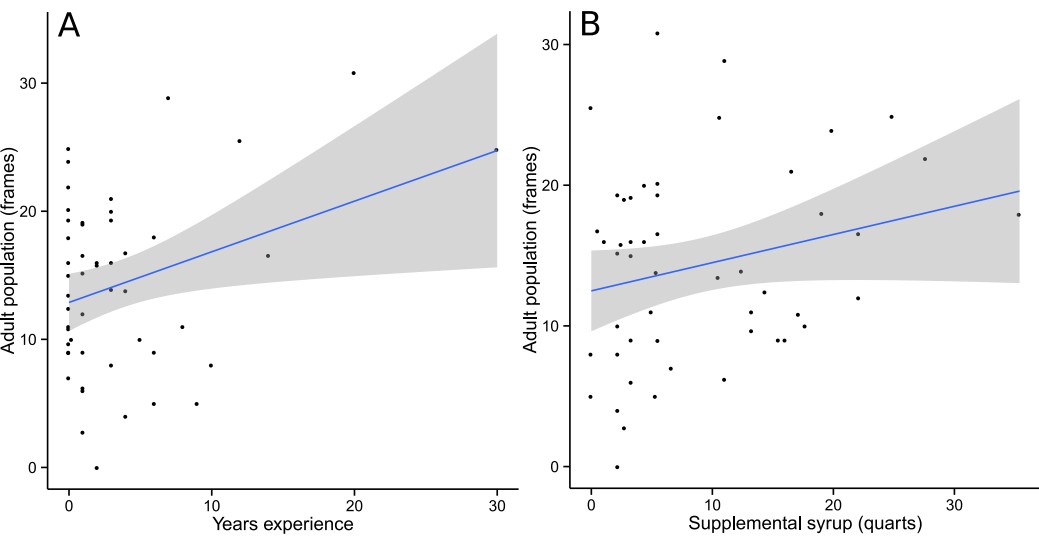

**Figure 4** **Adult population was positively correlated with beekeeper years of experience (A) and supplemental syrup feeding (B).** A 95% confidence band is shaded in gray.

## Modeling colony success metrics by urban landcover

In the subset of sites for which Urban + Crop was greater than 50% of total land cover, we found a significant ($p < 0.05$) negative relationship between *Food* and *Urban* (Fig. 5) at all spatial scales except for the two extremes of 0.5 km and 5 km; the relationship was strongest at the 2 km scale ($F = 6.041$, df $= 29$, $p = 0.02$). *Wax* was not significantly related to *Urban* ($p > 0.05$).

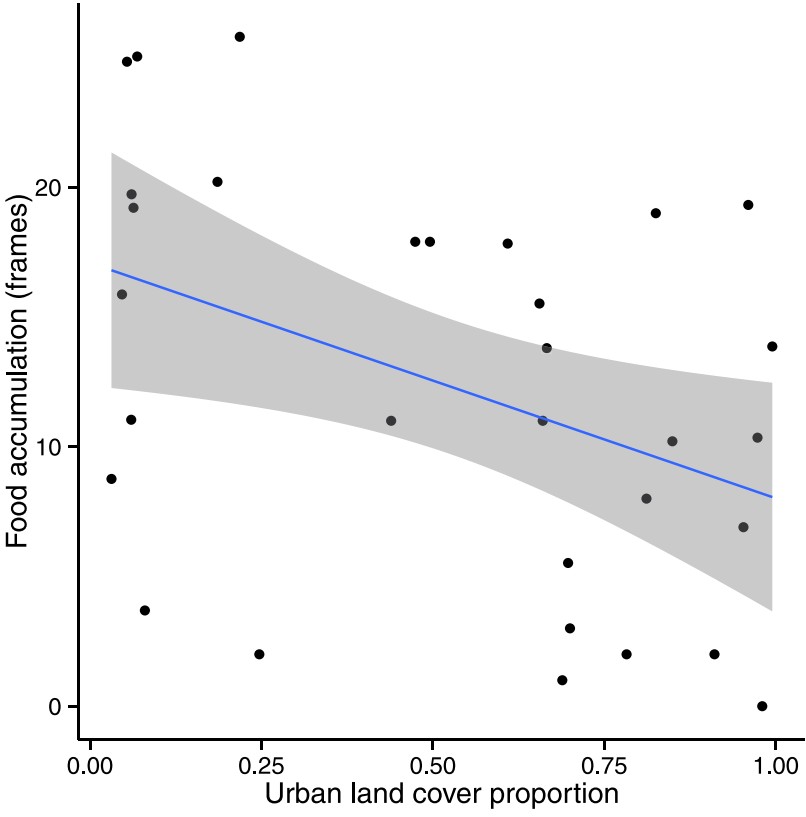

**Figure 5 Colony food accumulation decreased significantly with increasing urban land cover in sites where Urban + Crop > 50%.** This pattern was strongest at a 1 km radius (shown above). A 95% confidence band is shaded in gray.

## DISCUSSION

The negative responses of *Food* and *Wax* to PC2 indicate that food accumulation and wax production increase with surrounding cropland and decrease with forest/grassland. This finding seems to contradict the conventional wisdom that agricultural land conversion threatens honey bee nutrition through the depauperation of floral resources relative to semi-natural environments (*De La Rúa et al., 2009*), but is consistent with studies that have found honey bees to be notably resilient to natural habitat loss compared to other bee taxa (*Ricketts et al., 2008*; *Winfree et al., 2009*). The productivity of honey bees does not depend so much on the presence of undisturbed natural floral communities as it does on the availability of rich resources that can be exploited efficiently by cooperative foraging (*Visscher & Seeley, 1982*), and agricultural environments can offer honey bees surprisingly rich floral resources in the form of "weeds" (*Odoux et al., 2012*; *Requier et al., in press*). In Ohio, the largest honey yield is believed to come from non-native clovers (*Trifolium* spp. L.) (*Pellett, 1920*; *Bailey, 1955*; *Goltz, 1975*); these plants grow abundantly along roadsides, in field margins, and in grassy yards, but they are scarce in habitats shaded by forest canopy or dominated by the dense herbaceous vegetation of unmowed grassland. In addition to the clovers, *Erickson (1984)* observed that, under some conditions, honey bees will

forage very productively on soybean (*Glycine max* (L.) Merr.), and corn/soybean rotations comprise the vast majority of Ohio cropland. Dandelion (*Taraxacum officinale* FH Wigg.), one of the most important spring flora for honey bees in the Midwest (*Jaycox, 1976*) during the period of peak wax production, is distributed in much the same pattern as the clovers, thus favoring wax production in cropland over seminatural forest and grassland.

Interestingly, our finding that colony productivity is favored by cropland relative to forest/grassland is strikingly consistent with an anecdotal description of regional honey production in Ohio published nearly forty years ago (*Goltz, 1975*). In Goltz' account, the areas of "primary" and "secondary" importance for honey production are in the heavily cultivated glacial plains that comprise most of the state, while the forest-dominated Appalachian Plateaus in the southeast are described as only "marginally" productive.

The positive response of *AdultPop* to the management variables *experience* and *syrup* is difficult to interpret. In early spring, when new colonies are very small and limited in their foraging ability, it is standard practice to supplement colony nutrition with sugar syrup. All workers produced during the period of spring build-up, though, died long before colonies were inspected in the fall, so any positive effect of springtime management on adult population at time of inspection would have to be mediated by factors that allow colonies to increase reproduction later in the year. An alternative interpretation is plausible if we allow that significant feeding may have occurred later in the year. While supplemental feeding is normally concentrated in early spring, some Ohio beekeepers also feed their colonies in mid-late summer, a period of perceived dearth in natural forage. Feeding during the summer dearth period might trigger a population increase that would persist until fall inspection. Our survey did not distinguish between feeding at different times during the season. The effect of beekeeper experience on adult population is difficult to parse, as all aspects of hive management would be expected to improve with increasing experience. Somewhat ironically, a positive relationship between colony success and beekeeper experience might be explained by the tendency of more experienced beekeepers to perform less colony management; the enthusiasm of new beekeepers can lead to unnecessary interventions that do more to disturb natural colony function than to ameliorate ills (J Tew, pers. comm., 2014).

By late September and early October, when beekeepers were inspecting their colonies for the fall survey, the bees had likely already begun to reduce brood rearing in preparation for winter (*Graham, 1992*). This would explain the failure of both landscape and management variables in predicting *BroodPop*.

The negative relationship observed between *Food* and *Urban* in the subset of our sites strongly characterized by urban and/or agricultural land use does not support the popular opinion that urban landscapes favor honey bee success relative to agricultural landscapes. At least in Ohio, the relationship appears to be the opposite, and the fact that *Food* was the only success metric to respond to *Urban* ratio suggests a likely mechanism. The last major nectar and pollen flow in Ohio is usually from goldenrods (*Solidago* spp. L.) (*Morse, 1972*; DB Sponsler, 2014, unpublished data), which bloom prolifically from late summer into fall, roughly the same period during which beekeepers in our study were conducting

fall hive inspections and filling out the fall survey. At this time of year, honey bees rarely produce additional wax (*Lee & Winston, 1985*), and brood rearing has begun to slow down in preparation for winter (*Graham, 1992*), so incoming food is stored rather than being invested in brood or wax production. Goldenrods occur abundantly in uncultivated fields and conservation strips throughout agricultural landscapes, but they are relatively scarce in developed areas where vegetation is more often subject to mowing and weed control. This is consistent with the anecdotal observation of *Burgett, Caron & Ambrose (1978)* that urban hives tend to have poor late-season honey production, which the authors attribute to scarcity of late-blooming "weeds," including goldenrods.

We conclude that both landscape composition and colony management contribute to the success of nascent honey bee colonies in our study region. Due to complexities not explored in this study, the prediction of colony success was partitioned such that landscape predicted food accumulation and wax production, while colony management predicted only adult worker population. We find no support for the opinion that honey bees in urban landscapes are more successful than those in cropland. To the contrary, we find that colony food accumulation responds negatively to urban land cover in landscapes dominated by urban or agricultural land use, a pattern that we attribute to the influence of late-season floral availability, particularly goldenrods.

It is important to note that while model selection identified landscape composition as the best predictor of colony food and wax accumulation, the amount of unexplained variation in our models was high, indicating that factors other than the ones we measured are also at play in the determination of colony success. Such factors may include (1) fine-scale landscape variables that were not measurable using the NLCD dataset, (2) hive management variables not accounted for in our beekeeper survey, and (3) "in-hive" determinants of colony success like queen fertility, disease prevalence, and parasite load. We also suggest caution in generalizing our results beyond our study region. While the landscape of Ohio is broadly similar to much of the American Midwest, it would be premature to extend our findings to other ecoregions that may differ strongly both in natural flora and in agricultural practices.

## ACKNOWLEDGEMENTS

We sincerely thank the many Ohio beekeepers whose participation made this project possible. C Hoy and M Gardiner provided helpful statistical consultation, and JO Quijia-Pillajo assisted in GIS analysis.

### Funding

This study was funded by the Ohio State Beekeepers' Association. The funders had no role in study design, data collection and analysis, decision to publish, or preparation of the manuscript.

## Grant Disclosures

The following grant information was disclosed by the authors:
Ohio State Beekeepers' Association.

## Competing Interests

The authors declare there are no competing interests.

## Author Contributions

- DB Sponsler conceived and designed the experiments, performed the experiments, analyzed the data, contributed reagents/materials/analysis tools, wrote the paper, prepared figures and/or tables.
- RM Johnson conceived and designed the experiments, performed the experiments, analyzed the data, contributed reagents/materials/analysis tools, reviewed drafts of the paper.

## Human Ethics

The following information was supplied relating to ethical approvals (i.e., approving body and any reference numbers):

1. The Ohio State University Office of Responsible Research Practices.
2. Approval numbers: Protocol # 2012E0136, Protocol # 2013E0012.

## Supplemental Information

Supplemental information for this article can be found online at http://dx.doi.org/10.7717/peerj.838#supplemental-information.

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
