# Peer review of "Honey bee success predicted by landscape composition in Ohio, USA"

_PeerJ, doi:10.7717/peerj.838_

## Round 0.1 · original submission · Major Revisions

Please carefully consider all comments by the reviewers before resubmission. Please pay special care to interpretation of your results, and to put your work into the context of other published work.

Also, please consult the journal's policy on data sharing: https://peerj.com/about/policies-and-procedures/#data-materials-sharing, and take appropriate action.

·

Basic reporting

The paper is well written and provides information on the relationship between honeybee colony growth and landscape context. My main caveat is that most results present weak correlations, and probably this should be discussed in the paper. See detailed comments below.

Experimental design

The experimental design is clear, and only needs a few clarifications:

Line 85: "only one study hive": Did they choose a random one? Can be biased to report the best one? If so, volunteers with more hives, will tend to do better. Number of hives can may also be correlated with years of experience.
Line 87: "irreconcilable inconsistencies": May be remove "irreconcilable"?
Line 90: Are the sites independent? i.e. at least 5km apart one from each other. This is important.
Line 97: "Hadd = honey frames added to the hive prior to inspection". For non bee keepers, can you briefly explain how common is this practice and why is done? Same for Badd = brood frames added to the hive prior to inspection,
Line 137: can you report the correlation between success metrics? Would be nice to see if they correlate, or not.
Line 139: I don't understand the word "coupled" in this context.
Line 140: what "a year-only" model means? Is this referring to the years of experience variable? If so, I understand this model is part of the "all combinations of ..." and do not need especial mention.

Validity of the findings

I can't find where the raw data is deposited.

The results are nicely presented, but I think the conclusions should be tone down a bit. Despite the best model indicates landscape is important, the models has a lot of unexplained variance. My interpretation of the data is that e.g. crops tend to be good habitats, but that there is a huge variability among landscapes dominated by crops, and other factors like flower quality may be more important to look at.

Line 170: You can use Akainke weighs to show PC2 is retained in most models.
Line 173: If you use a AIC approach I wouldn't report p-values. Otherwise, you should correct for the multiple comparisons you are doing. With so many models you will find some variables significant just by chance.
Line 181: Why you need to log-transform a ratio?

Another thing is that would be elegant to select the radius a priori, to minimise the amount of tests done, but I guess is to late for that.

Additional comments

The article reads good, but given your data is from a few localities in the same region, and the adjusted r's are very low, I would make sure the reader gets the idea that "coarse habitat composition is important, BUT there is a lot of variability within similar habitat compositions".

Some in line comments:
In abstract: I would remove the "we are aware of no published studies" which is unnecessary, specially for this Journal.
Line 83: Remove "be"
Line 210: Sorry this comment is out of place, but I just though: Is amount of syrup per hive? or per site? If per site, should be corrected by number of hives in a site.
Line 225: Alternatively, you may find little effects on those variables because the measure is quite coarse (not quantitative). I think this deserves a mention.
Table 1: Note the intercept only model is almost as good as any other model (AIC +0.87). This, along the low adjusted r's deserves a mention in text.

Reviewer 2 ·

Basic reporting

This study uses beekeeper survey data (n=50 completed surveys from two seasons) to relate measures of honey bee colony success to surrounding land use in Ohio, USA. Given the predicted importance of landscape composition for honey bee colony development, I agree with the authors that this has been studied surprisingly little.

There is, however, major room for improvements when it comes to putting this study into the context of other recently published work. The first study in the example list below of references not taken into account is critical to incorporate because it to my understanding is a study that “quantitatively measure colony success in response to local landscape variables”:

Odoux, J. F., Aupinel, P., Gateff, S., Requier, F., Henry, M., & Bretagnolle, V. (2014). ECOBEE: a tool for long-term honey bee colony monitoring at the landscape scale in West European intensive agroecosystems. Journal of Apicultural Research, 53, 57-66

Garbuzov, Mihail, Roger Schürch, and Francis LW Ratnieks. Eating locally: dance decoding demonstrates that urban honey bees in Brighton, UK, forage mainly in the surrounding urban area. Urban Ecosystems (2014): 1-8.

Donkersley, P., Rhodes, G., Pickup, R. W., Jones, K. C., & Wilson, K. (2014). Honeybee nutrition is linked to landscape composition. Ecology and evolution, 4(21), 4195-4206.

Requier, F., Odoux, J. F., Tamic, T., Moreau, N., Henry, M., Decourtye, A., & Bretagnolle, V. (2014). Honey bee diet in intensive farmland habitats reveals an unexpectedly high flower richness and a major role of weeds. Ecological Applications.

Experimental design

L59-64 How did you identify beekeepers / how many were asked to participate?

L75-77 What was the minimum, mean and maximum distance between hives participating?

L86-90. It would be useful to separate discarded surveys between the categories mentioned (missing data / inconsistent / hive died).

Validity of the findings

Important details are missing and some clarifications are needed in order to fully assess the validity of the findings (see detailed comments below).

L75-77 Were any locations or beekeepers identical in the two years, and if so how did handle that in analyses?

L100. Please check and clarify equation for net wax production. What is a “drawn but mostly empty frame”? Why – H(add), but + B(add) and + D(add)?

L106. Did you consider any effects of B(rm) and B(add) on BroodPop?

L134-156. How did you verify model assumptions, for example the error distribution of residuals?

L145-156. The approach to analyse the ratio between two land cover classes for a subset of the data seems unusual and further justification is needed. The results of this analysis is not discussed in relation to the finding that PC1 (urban - rural axis) does not seem to be important.

L146-147 What was the sample size of this subset?

L160-167. With these results at hand, I wonder whether it would be possible to analyse directly the effect of proportion urban land and proportion cropland on your response variables (having in mind the correlations between land cover classes)? What do you gain by doing a PCA?

L174. What are the test statistics and degrees of freedom for the p-values reported - this applies throughout.

L185-248. Discussion in general: Nice discussion about potential explanations, but specific to Ohio. Note that the question about the role of landscape composition for honey bee colony performance is generally stated in the introduction. Do you think that the results are generally applicable – why/why not?

L195-197, L201-202, L236-238. Are these personal observations?

L210. How do you explain the effect of “years” on AdultPop?

Additional comments

L22. / Introduction in general: What is “colony success” / how do you define it?

L109. Suggested change of name for variable “years” as you also have another variable called “year”.

Table 1. How was deltaAICc calculated for intercept only / year only models?

L224. This can explain

L286. Check reference title.

Figure legends 3 and 5. I recommend to not speculate on results in figure legends.

---

## Round 0.2 · accepted · Accept

I think you have done a really good job responding to the comments by the referees and improving the manuscript. I find it acceptable for publication in its present form.